# IR Study on Cellulose with the Varied Moisture Contents: Insight into the Supramolecular Structure

**DOI:** 10.3390/ma13204573

**Published:** 2020-10-14

**Authors:** Stefan Cichosz, Anna Masek

**Affiliations:** Institute of Polymer and Dye Technology, Faculty of Chemistry, Lodz University of Technology, Stefanowskiego 12/16, 90–924 Lodz, Poland; stefan.cichosz@p.lodz.pl

**Keywords:** cellulose, supramolecular structure, moisture content, IR spectroscopy, hydrogen bonding

## Abstract

The following article is the first attempt to investigate the supramolecular structure of cellulose with the varied moisture content by the means of Fourier-transform and near infrared spectroscopy techniques. Moreover, authors aimed at the detailed and precise presentation of IR spectra interpretation approach in order to create a reliable guideline for other researchers. On the basis of obtained data, factors indicating biopolymer crystallinity and development of hydrogen interactions were calculated and the peaks representing hydrogen bonding (7500–6000 cm^−1^, 3700–3000 cm^−1^, and 1750–1550 cm^−1^) were resolved using the Gaussian distribution function. Then, the deconvoluted signals have been assigned to the specific interactions occurring at the supramolecular level and the hydrogen bond length, as well bonding-energy were established. Furthermore, not only was the water molecules adsorption observed, but also the possibility of the 3OH⋯O5 intramolecular hydrogen bond shortening in the wet state was found-from (27,786 ± 2) 10^−5^ nm to (27,770 ± 5) 10^−5^ nm. Additionally, it was proposed that some deconvoluted signals from the region of 3000–2750 cm^−1^ might be assigned to the hydroxyl group-incorporated hydrogen bonding, which is, undoubtedly, a scientific novelty as the peak was not resolved before.

## 1. Introduction

This article debates on the cellulose structure changes which might possibly occur during the moisture absorption/desorption processes and how they affect the infrared spectra. The inspiration to carry out this research study was the fact reported in the previous years, namely, while cellulose is subjected to the multiple wetting-drying cycles, the biopolymer changes its structure and properties [1,2,3].

Nonetheless, in the past, it was barely explained what is actually happening with the cellulose fibers during the moisture absorption/desorption processes. Most often, only the native and fully wet/dried states were described [4,5]. Therefore, the aim of this research was to investigate the biopolymer chemical structure and hydrogen interactions at some certain water content levels during the moisture absorption process.

The mentioned above phenomenon, taking place while cellulose is alternately wetted and dried, is called ‘hornification’ and it could be explained with the hydrogen bonds reorganization favored by some conformational changes in cellulose chains [4,6].

Nevertheless, some studies refer that this process may also lead to the creation of new chemical bonds, e.g., lactone ones [4,7], as the biopolymer swelling decreases when the hornification process is performed. According to the authors, the additional chemical bonds between the cellulose macromolecules are able to create the specific net and, therefore, prohibit biopolymer swelling and dissolution [4].

In spite of the fact that cellulose is most often referred as a material of a highly polar nature and the structure changes taking place during the hornification process are usually explained with −OH interactions [8,9,10], the estimated cellulose oligomers stacking association energy of hydrophobic pairing turns out to exhibit higher values, when compared with the hydrogen bond contribution [11].

Therefore, cellulose is advised to be regarded as the amphiphilic material and it should not be considered only in terms of interactions characteristic of the polar substances [12].

The problem of cellulose amphiphilicity has been widely described regarding the dissolution possibilities of the discussed biopolymer [12,13,14]. Cellulose is insoluble in water and in typical organic solvents. Yet, it is possible to dissolve the biopolymer in some particular liquids, which, according to the current state of knowledge, are the systems of the totally different properties [12].

Of course, it is not surprising that cellulose is insoluble in non-polar solvents, as the biopolymer macromolecule possesses some hydroxyl groups with the ability of the hydrogen bonds creation [15], but it also cannot be dissolved in polar solvents [16]. This is unexpected due to the fact that glucose, the primary building unit of cellulose, may be dissolved in aqueous media with ease [12]. What is more, also some glucose derivatives, being hydrophobized, exhibit good solubility in water [12,17,18].

Analyzing the cellulose chemical composition a little deeper, it could be noticed that the amount of −OH moieties with protons able to form hydrogen bonds is lesser than the number of oxygen atoms that are capable of creating this kind of interaction [14]. Consequently, there are many possibilities for, e.g., water molecules, to form some additional hydrogen bonds with cellulose (reason for high hygroscopicity).

Concluding, cellulose insolubility in the polar solvents explained only on the basis of very strong intermolecular hydrogen bonds cannot be correct, as carbohydrate-carbohydrate hydrogen bonding is not the only interaction possible during the biopolymer dissolution [19].

Considering the given above example of cellulose interactions observed during the dissolution attempts, discussed biopolymer should be foreseen as the amphiphilic material. Furthermore, thinking about the macromolecular compounds, even a slight increase in the amphiphilic properties may fully change the behavior of the whole material [12,14]. Therefore, understanding of different interactions present in the system is crucial for appropriate characterization of the material, e.g., during moisture absorption/desorption experiment. Moreover, the amphiphilicity is not only observed in case of the discussed biopolymer, but also for, e.g., poly(ethylene glycol) (PEG) [20].

Moving on to the cellulose supramolecular structure, it may be described by a two-phase model with regions of high (crystalline) and low (amorphous) orientation of macromolecule chains [21,22]. Due to the further investigation of the crystalline regions, the amphiphilic character of cellulose becomes even more complex-the chain of glucose rings has ability to exhibit sides of a varied polarity depending on the considered direction [12].

While giving a closer look at cellulose crystals [23], it could be observed that in the equatorial direction of glucopyranose ring, cellulose exhibit a hydrophilic nature. Then, all three −OH moieties are located on the equatorial positions of the ring. However, the axial direction is highly hydrophobic as hydrogen atoms of C-H bonds are located on the axial positions [12].

Gathering all the information given above, cellulose macromolecules exhibit structural anisotropy. Furthermore, as a consequence of intra- and intermolecular hydrogen bonds, they form flat ribbons with the sides highly varying in the polarity [12,24,25]. It is an incredibly crucial information regarding the possibility of cellulose properties control with the appropriate organization of hydrogen bonding at the supramolecular level [26,27].

Therefore, this article concentrates on the hydrogen bonds formation described as a function of the water content in cellulose. What should be emphasized, this kind of research approach is introduced for the first time and successfully fills the gap in the current state of knowledge-till now, only the dry and wet cellulose states were fully described. Moreover, this article provides a new point of view regarding the development of hydrogen bonding during the moisture absorption/desorption process and the presented analysis was performed with the employment of commonly used infrared spectroscopy methods, namely: Fourier-transform infrared spectroscopy (FT-IR), near infrared spectroscopy (NIR), which are simple in exploitation and data analysis.

## 2. Materials and Methods

### 2.1. Materials

The Arbocel^®^ UFC100 Ultrafine Cellulose for Paper and Board Coating (UFC100) from J. Rettenmaier & Soehne (Rosenberg, Germany) was employed in all performed experiments. It is in a powder form (density: 1.3 g/cm^3^, average fiber length: 6–12 μm, pH: 5.0–7.5). Cellulose is insoluble in water and fats, but it exhibits a high water binding capacity.

Moreover, phosphorus oxide (V) (Chempur, Piekary Slaskie, Poland) was employed in a role of a desiccant cartridge in a desiccator during the cellulose conditioning before the further experiment. The mentioned oxide is in a solid state, exhibits pH of approximately 1.5 and density of about 2.3 g/cm^3^. Phosphorus oxide (V) is referred to provide an atmosphere of the relative humidity (RH) of approximately 0% [28]. 

Then, the moisture absorption process has been carried out in another desiccator filled with the saturated aqueous solution of potassium nitrate (Chempur, Piekary Slaskie, Poland). The solubility of the salt in water is 316 g/L (20 °C). According to literature, such solution should provide the atmosphere of a specific relative humidity which is equal about 96% [28].

### 2.2. Preparation of Cellulose Samples

Before the cellulose conditioning process the material has been divided into separate weighing bottles with 0.1 g (Fourier-transform infrared spectroscopy, near infrared spectroscopy), either 1.5 g (Karl Fischer titration) of cellulose in each vessel (separate weighing bottle for each separate FT-IR/NIR/titration experiment). Further preparation of the specimens has been divided into three stages:cellulose conditioning in the desiccator filled with phosphorus oxide (RH = 0%) for 7 days (all prepared weighing bottles with cellulose; 35 × 70 mm);moisture absorption experiment: desiccator filled with saturated solution of potassium nitrate; experiment lasted 24 h (all prepared weighing bottles with cellulose; part of the specimens prepared for desorption experiment was put into the separate, twin desiccator filled with the same solution); measurements after: 0, 1, 3, 5, 8, 24 h (the last measurement after 24 h is the first measurement of the moisture desorption stage at 0 h);moisture desorption experiment: laboratory dryer (Binder, Tuttlingen, Germany) at 100 °C for 8 h (only weighing bottles prepared for desorption experiment); measurements after: 0, 1, 3, 5, 8 h.

In order to obtain reliable results, the described above process of moisture absorption desorption was repeated 5 times (5 samples). Then, the average values and standard deviations were calculated.

The graph in Figure 1 reveals the changes in the average moisture content (measured with Karl Fischer titration; further description in Section 2.3.3) during the moisture absorption/desorption experiment.

Throughout the article, all analyzed parameters are presented as a function of moisture content and not time. Therefore, taking into consideration the fact that the values of water content for desorption stage are similar, from 1–8 h of the carried out experiment (desorption stage), only the last measurement of this segment is considered in the further investigation (after 8 h of cellulose sample drying). The considered points are marked with the blue circles in Figure 1.

Moreover, exemplary spectra are shown for 3 points reflecting different states of cellulose: 1st point-before the experiment, 2nd point-after 24 h of moisture absorption, 3rd point-dried cellulose.

### 2.3. Methods

#### 2.3.1. Fourier-Transform Infrared Spectroscopy (FT-IR)

Fourier transform infrared spectroscopy (FT-IR) absorbance spectra has been investigated within the 4000–400 cm^−1^ range (64 scans, 4 cm^−1^ resolution, absorption mode). The experiment has been performed with the use of Thermo Scientific Nicolet 6700 FT-IR spectrometer (Thermo Fischer Scientific Instruments, Waltham, MA, USA) equipped with diamond Smart Orbit ATR sampling accessory.

Moreover, the following parameters have been calculated: total crystalline index (TCI), lateral order index (LOI), hydrogen bond intensity (HBI), as a ratio between the peaks intensities, respectively: 1370–1360/2900–2890 cm^−1^, 1430–1420/897 cm^−1^, 3340–3330/1315 cm^−1^ (depending on the shifts between the peaks during the moisture absorption/desorption).

Furthermore, some parts of the FT-IR spectra, namely: 3700–3000 cm^−1^, 3000–2750 cm^−1^, 1750–1550 cm^−1^, were resolved by using the Gaussian distribution function into, respectively, 3, 5 and 2 signals. Absorbance of the band obtained from a local baseline between adjacent valleys was automatically calculated at the maximum absorbance found by OriginPro 2020 software.

For deconvoluted peaks which are assigned to the intra- and intermolecular, as well as cellulose-water hydrogen bonds, the bond energy (*E_H_*) and its length (*R_H_*) have been calculated according to the equations given below:(1)$$E_{H}=\frac{1}{k}\frac{\nu_{0}-\nu }{\nu_{0}}\;\left[kJ\right]$$
(2)$$R_{H}=\left(2.84-\frac{\nu_{0}-\nu }{4430}\right)\cdot 0.1\;\left[\mathrm{nm}\right]$$
where:
$\nu_{0}$ standard frequency of free OH groups ($\nu_{0}$ = 3600 cm^−1^)ν frequency of the bonded −OH groups [cm^−1^]k constant ($\frac{1}{k}$ = 262.5 kJ)

#### 2.3.2. Near Infrared Spectroscopy (NIR)

In this method, the near-infrared region of the electromagnetic spectrum was used. The measurements were carried out in the range of 10,000–4000 cm^−1^ in absorption mode—64 scans, 4 cm^−1^ resolution (Thermo Fischer Scientific Instruments, Nicolet 6700, Waltham, MA, USA).

Furthermore, some parts of the NIR spectra, namely 7500–6000 cm^−1^, were resolved by using the Gaussian distribution function into 6 signals. Absorbance of the band obtained from a local baseline between adjacent valleys was automatically calculated at the maximum absorbance found by OriginPro 2020 software.

#### 2.3.3. Karl Fischer Titration (KFT)

Fischer titration was done with the use of TitroLine Alpha (Schott, Mainz, Germany) device. The measurement has been carried out with the use of Hydranal Solvent E and Hydranal Titrant 5E supplied by HoneywellonHon Fluka (Loughborough, UK). For each experiment approximately 1.3 g of cellulose sample and 30 mL of Hydranal Solvent E were taken. The water content in the natural filler has been established as follows [29]:(3)$$BI_{2}+BSO_{2}+B+H_{2}O\to 2BH^{+}I^{-}+BSO_{3}$$
(4)$$BSO_{3}+ROH\to BH^{+}ROSO_{3}{}^{-}$$

The anode solution consists of an alcohol (*ROH*), a base (*B*), *SO*_2_ and *I*_2_. The Pt anode generates *I*_2_ when current is provided through the electric circuit. The net reaction as shown above is oxidation of *SO*_2_ by *I*_2_. One mole of *I*_2_ is consumed for each mole of *H*_2_*O*. In other words, 2 moles of electrons are consumed per mole of water.

## 3. Results and Discussions

### 3.1. Fourier-Transform Infrared Spectroscopy Analysis

#### 3.1.1. General Description of the Spectra

The precise and deep analysis of the spectrum combined with information gathered in literature enables the distinguishing of the signals in a very specific way. The particular carbon/oxygen atom may be assigned to the appropriate peaks visible in the spectrum. The outcome of this analysis is presented in both Figure 2 and Table 1.

From the cellulose supramolecular structure point of view, which is the subject of this article, the most interesting spectrum parts are: between 3700 and 3000 cm^−1^ (where the hydrogen bond formation could be observed), from 1420–1430 cm^−1^ (associated with the amount of crystalline structure of the cellulose) and in the region of 900–890 cm^−1^ (assigned to the amorphous region) [30].

Moreover, it is highly advised to be familiar with the crystalline and amorphous regions spectral displacement as the ratio between the peaks assigned to these two cellulose states, namely 1430–1420/897 cm^−1^, may indicate some information about the crystallinity degree of cellulose and is referred as lateral order index (LOI) [31]. Additionally, the ratio between the peaks at ~1370 cm^−1^ and 2900–2890 cm^−1^ is regarded as total crystalline index (TCI) [31]. Both of these parameters are very useful in cellulose properties description and reliably correspond with the crystalline index calculated according to the X-ray diffraction (XRD) experiment results [32].

Moreover, the results from the fingerprint region in the range from 1400–900 cm^−1^ should be in good agreement with the changes observed in the broad band at 3700–300 cm^−1^, corresponding to the strong OH stretching and flexing vibration frequencies of the intra- and intermolecular hydrogen bonds of cellulose [25].

Crystalline structures of cellulose are understood as regions in which the high order of cellulose macromolecules is evidenced. The high order, in turn, is directly connected with some specific interactions present within cellulosic materials-the hydrogen bonds. Their patterns are considered as one of the most influential factors in determining the structure and properties of the discussed biopolymer [25]. Some possibilities of their formation are presented in Figure 3. Intramolecular hydrogen bonds could be created between 3OH⋯O5 and 2OH⋯O6 (Figure 3a). On the other hand, intermolecular hydrogen bonds, binding two or many macromolecules together, are mostly formed in native cellulose between 6OH⋯O3′ (Figure 3b) [24,32]. 

Furthermore, as it was mentioned in Section 1, the amount of hydroxyl groups with protons capable of forming hydrogen bonds is actually less than the number of oxygen atoms that are capable to form this kind of interaction [14]. Therefore, there are many possibilities for a liquid like water (it poses a proton capable of hydrogen bond) to create additional physical interactions with the cellulose molecules. This is presented in Figure 3c.

Additionally, in Figure 3d the carbon and oxygen atoms numbering has been presented in order to enable easy reading of the whole scheme.

Evaluating the hydrogen bonding development, a particular factor may be calculated, namely hydrogen bond intensity (HBI). It is referred as a ratio between the peaks at 3400–3000 cm^−1^ and 1320–1310 cm^−1^ (depending on the wavenumber shifts) [31]. The HBI of cellulose is closely related to the crystal system and the degree of intermolecular regularity (crystallinity). Accordingly, it might provide some specific information on cellulose water content [32]. 

Nevertheless, considering the absorbed moisture amount observation, the peak at approximately 1640 cm^−1^ is more appropriate to analyze, as it is assigned to H-O-H angle vibrations in water molecules [35].

In Figure 4a the cellulose of different moisture content spectra, obtained during the water absorption/desorption experiments, are revealed. The most interesting and clearly visible changes are marked: development of intra- and intermolecular hydrogen bonds between 3700–3000 cm^−1^ and water absorption at around 1640 cm^−1^ (further investigation in Section 3.1.2).

In turn, regarding the described crystalline/amorphous regions, it is troublesome to observe some specific changes without deeper spectra investigation. While the TCI, LOI, HBI factors are calculated, some variations could be detected (Figure 4b). It is visible that with the increasing moisture content in cellulose sample, the TCI slightly decreases (drops from 1.39 ± 0.03 to 1.31 ± 0.04) and HBI raises at the same time. This may indicate the loosening of the cellulose structure (crystallinity decreases) with simultaneously ongoing development of hydrogen bonds. 

What is important, the carbohydrate-water molecule interactions should be taken into account and considered regarding the trend detected for HBI-it changes from 1.31 ± 0.05 to 1.7 ± 0.1. Visible raise could be explained by the increasing amount of cellulose-water hydrogen bonds. Otherwise, if it was due to the cellulose-cellulose interactions, the TCI would also increase (higher and more stable order thanks to the hydrogen bonding) [38].

What is interesting, LOI remains almost unchanged (it varies from 0.56 ± 0.02 to 0.57 ± 0.01) and extremely differentiated (high error bars) among the repeated measurements. These differences between the both factors might be caused by the fact that LOI is correlated to the overall degree of order in cellulose, while TCI is referred to be proportional to the crystallinity degree. Both factors provide information about the order of macromolecules within the sample but in a slightly varied way [31]. 

Summarizing the gathered information, moisture content seems to highly affect the intensity of hydrogen bonds within the cellulose macromolecules causing their development. According to the literature, this may be expected that the biopolymer crystallinity degree would increase as the hydrogen bonds are known to favor the high order of the cellulose macromolecules [31,39].

However, the expected raise in the biopolymer crystallinity seems not to occur during the moisture absorption, even that the hydrogen bonding intensity increases. This phenomenon is probably observed due to the fact that the hydrogen bonding becomes more complex regarding the carbohydrate-water hydrogen bonding and not carbohydrate-carbohydrate interactions. Therefore, the increasing HBI factor may be misleading-while giving a closer look into literature, water molecules penetrating entangled cellulose chains are referred to loosen the biopolymer structure contributing to its swelling and increasing the distances between the macromolecules rather than raising the amount of crystalline, high-ordered structures [4,6].

Therefore, all of the investigated factors need to be carefully considered regarding the possible changes in the cellulose structure. Nevertheless, the water presence influence on the hydrogen bonding development has to be taken into account in order to avoid false data interpretation. It is hard to distinguish the effect of cellulose-water interactions from cellulose-cellulose intra- and intermolecular hydrogen bonds.

#### 3.1.2. Hydrogen Bonding Investigation

According to the information presented in Figure 3a–c different hydrogen bonds within the cellulose sample might be evidenced: intramolecular (3OH⋯O5 and 2OH⋯O6), intermolecular (6OH⋯O3′), as well as hydrogen bonds with water molecules adsorbed/absorbed by the cellulose fibers. The overall development of hydrogen bonding within a cellulose sample may be investigated with, e.g., HBI factor, shape and the intensity of some specific absorption bands (e.g., 1750–1550 cm^−1^, 3700–3000 cm^−1^).

However, observation of particular hydrogen bonds is impossible without the further mathematical operations carried out on the spectra. The spectrum could be deconvoluted either the first/second derivative might be calculated [24,40]. These operations enable observation of particular signals that form together the peak visible in the spectrum (resolving of the spectra). In further sections, the deconvoluted peaks assigned to the absorbed water and inter/intramolecular hydrogen bonds in cellulose (Section 3.1.2) are presented. Moreover, in the Section 3.1.3, an interesting observation is introduced-the peak between 3000–2750 cm^−1^ have been deconvoluted and analyzed for the first time.

Depending on the wavelength of the analyzed deconvoluted peak assigned to the particular hydrogen bond, its energy and distance could be calculated [31].

##### Deconvolution of the Peak between 1750–1550 cm^−1^

According to literature [35], moisture causes the occurrence of multiple peaks at following wavenumbers: 700 cm^−1^ (out-of-plane vibrations of O-H groups or rotational vibrations of the whole water molecule), 1640 cm^−1^ (H-O-H angle vibrations), 2100 cm^−1^ (vibrations from the scission and rocking of water).

However, as it was mentioned before, the absorption band at approximately 1640 cm^−1^ seems to be very interesting regarding the opportunity of the cellulose moisture content analysis [35]. Moreover, it is usually well-visible considering the cellulose spectrum.

In Figure 5a,b it is presented that the discussed peak between 1750–1550 cm^−1^ could be resolved into two signals both of which, according to literature, could be assigned to hydrogen bonding between cellulose and water [24,32]. Nonetheless, it is not certain what is the difference between these two deconvoluted peaks and what type of hydrogen bond they reflect. Moreover, it was not possible to observe a peak in the spectra for the cellulose water content lower than 1 wt%. This was shown before in Figure 4a.

In the carried out research, it could be observed for the first time, how the area, wavenumber and absorbance of the deconvoluted signals change with the raising moisture content (Figure 5c,d). It is easily seen that peak no. 1 (sharp and high) is not shifting its wavenumber with the absorbance and area variations during the water absorption [dry state-(1650 ± 10) cm^−1^; wet state-(1644.6 ± 0.6) cm^−1^]. However, an opposite situation might be detected in case of the peak no. 2 (flat and wide)—its wavenumber shifts significantly from (1638 ± 7) cm^−1^ to lower values with the increasing moisture content. An interesting observation might be that the area and absorbance changes seems to exhibit the same trend regarding both analyzed signals.

Considering Figure 5e,f, some variations in energy of cellulose-water hydrogen bonding and its length could be detected. The changes are more intense for peak no. 2, as the energy of the bond and its distance are dependent on the signal wavenumber-the energy of the bond raises significantly from (143.0 ± 0.5) kJ to (145 ± 1) kJ with the increasing moisture content. At the same time, a drop in the hydrogen bond length from (23986 ± 1) 10^−5^ nm to (2391 ± 3) 10^−4^ nm is detected. Contrary, changes of the peak no. 1 are less intense-the bonding energy is in the region of (142.3 ± 0.5) kJ. It is not surprising, as the peak does not shift much along the X axis and it stays almost constant during the whole experiment.

Taking into consideration gathered information, it is possible to detect two separate signals originating from cellulose-water hydrogen bonding which has been reported in the literature before [32]. Moreover, it was shown that the peak no. 1 and peak no. 2 reveal similar trends regarding the absorbance and area variations. However, considering the wavenumber changes, peak no. 1 does not shift significantly along the X axis in contrast with the peak no. 2 which moves toward lower wavenumbers.

##### Deconvolution of the Peak between 3700–3000 cm^−1^

Undoubtedly, water molecules absorbed by cellulose fibers highly influence the properties of the biopolymer [41]. Nevertheless, these are the cellulose-cellulose intra- and intermolecular bonds which are crucial for determining the characteristics of the discussed material [32]. 

Therefore, the most important deconvolution is presented in this section, as the region 3700–3000 cm^−1^ is assigned to the hydroxyl groups present in the biopolymer structure [32,38]. In Figure 6a–c the resolved peak divided into three separate signals is presented. Peak no. 1 and peak no. 2 are assigned to, respectively, intramolecular 3OH⋯O5 and 2OH⋯O6 interactions, while peak no. 3 is attributed to the intermolecular 6OH⋯O3′ hydrogen bond. Additionally, it was accepted in literature that in the structure of native cellulose, intramolecular hydrogen bonds of the following types: 3OH⋯O5 and 2OH⋯O6 are present on both sides of the chain [25]. Moreover, also another theory has been proposed-the peak shifted the most to the higher wavenumber might be described as the shoulder of the large OH peak of the water originating from stretching bands belonging to OH groups engaged in hydrogen bonding. Moisture visible in this region could be considered as a more loosely bound water-indirectly via another water molecule [35]. The hydrogen bonding energy variations are revealed in Figure 7.

After a short analysis, it could be noted that the deconvoluted signals do not change significantly during the moisture absorption as the system of hydrogen bonds in cellulose fibers is quite stable [34]. Nevertheless, some variations between the graphs shown in Figure 6a–c could be observed. 

To begin with, the raise in the intensity and area of the peaks with the increasing water content could be observed for all of investigated materials (Figure 6d–f). Moreover, the detected trend is similar throughout different specimens. According to literature, the observed peak intensity/area rise might be caused by both water content itself or the development of cellulose intra- and intermolecular hydrogen bonds [42]. 

Moreover, some wavenumber shifts between the analyzed signals could be detected: peak no. 1- from (3455.2 ± 0.4) cm^−1^ to (3480 ± 15) cm^−1^, peak no. 2- from (3328.2 ± 0.8) cm^−1^ to (3320 ± 5) cm^−1^, peak no. 3- from (3214 ± 6) cm^−1^ to (3230 ± 10) cm^−1^. This kind of variations are referred to be less influenced by the spectral intensity [43] and, therefore, more reliable to extract important information. 

Peak assigned to 3OH⋯O5 intramolecular hydrogen bond shifts in the direction of lower wavenumber, while the peaks attributed to 6OH⋯O3′ intermolecular hydrogen bond and 2OH⋯O6 intramolecular hydrogen bond move toward higher wavenumber. This may indicate some energy and bond length variations between the analyzed hydrogen interactions.

Moving forward to Figure 7a–c, which reveals the calculated energy and the length of hydrogen bonds, it could be noticed that, actually, 3OH⋯O5 intramolecular hydrogen bond energy raises from (19.93 ± 0.8) kJ to (20.4 ± 0.4) kJ, while energy of the 6OH⋯O3′ intermolecular hydrogen bond and 2OH⋯O6 intermolecular hydrogen bond slightly drops, respectively, from (27.9 ± 0.1) kJ and (10.40 ± 0.3) kJ to (27 ± 1) kJ and (9 ± 1) kJ. Increased or decreased hydrogen bond energy might be related to the raise of the number of hydrogen bonds in the system and, consequently, it may indicate some changes in the crystalline region [25].

Consequently, according to the calculated energy changes, 3OH⋯O5 intramolecular hydrogen bond becomes shorter, while the other two hydrogen bonds are getting longer. This may cause the occurrence of the tension within the material and could indicate less ordered packing arrangement of the macromolecule segments. The distances of observed hydrogen bonds differ from the values detected for native cellulose fibers—lengths of the 3OH⋯O5, 2OH⋯O6 and 6OH⋯O3′ hydrogen bonds in native cellulose are referred to be, respectively, 0.278 nm, 0.287 nm, 0.279 nm [25]. In the following research they vary: from (27,786 ± 2) 10^−5^ nm to (27,770 ± 5) 10^−5^ nm for 3OH⋯O5, from (28,073 ± 1) 10^−5^ nm to (28,120 ± 40) 10^−5^ nm for 2OH⋯O6, from (2753 ± 1) 10^−5^ nm to (27,570 ± 50) 10^−5^ nm for 6OH⋯O3′.

On the basis of the carried out research, it could be concluded that there is a possibility of 3OH⋯O5 intramolecular hydrogen bond shortening while cellulose is subjected to moisture. At the same time, the other two hydrogen bonds (2OH⋯O6 and 6OH⋯O3′) may elongate changing the tension within the cellulosic material and loosening the fiber molecular packing. Nevertheless, this phenomenon should be investigated more deeply in the future with the employment of different techniques.

#### 3.1.3. Deconvolution of the Peak between 3000–2750 cm^−1^

Furthermore, an interesting case is revealed in Figure 8. In literature it is often underlined that the peak between 3000–2750 cm^−1^ might be highly affected by the water content in cellulose sample. Nevertheless, it was never deconvoluted into separate signals.

Some studies assign this absorption band only to C-H bonds typical to alkyl chains [42,44] and only in the researches which debates about the moisture content in natural fibers, it is partially referred to be influenced by water content [24,32,34]. Moreover, an interesting paper from 1998 indicates the possibility of assigning one of the deconvoluted signals from this broad absorption band to −OH moieties [45].

Regarding the performed calculations, the absorption band between 3000–2750 cm^−1^ could be resolved into 5 separate signals (Figure 8a–c). It is the lowest number of peaks into which the broad absorption band might be successfully divided. Nevertheless, according to the current state of knowledge, it is troublesome to assign these deconvoluted signals to some particular chemical bonds present in the structure of cellulose. However, possibly, most of them could be attributed to different C-H bonds which are present in the structure of natural fiber [42,44]. 

Taking into consideration what is visible in Figure 8a–c and how the deconvoluted signals change during the moisture absorption process, it could be concluded that, indeed, water amount in cellulose sample highly affects the shape of the whole absorption band, as well as deconvoluted peaks. They shift significantly: peak no. 1- from (2960 ± 10) cm^−1^ to (3061 ± 30) cm^−1^, peak no. 2- from (2944.9 ± 0.5) cm^−1^ to (2968.2 ± 0.2) cm^−1^, peak no. 3- from (2900 ± 1) cm^−1^ to (2943 ± 1) cm^−1^, peak no. 4- from (2880 ± 10) cm^−1^ to (2865 ± 6) cm^−1^ and peak no. 5- from (2825 ± 2) cm^−1^ to (2864 ± 6) cm^−1^.

Moreover, the absorption bands in between 3700–3000 cm^−1^ and 3000–2750 cm^−1^ seem to merge with each other while the moisture content increases—the peak seems to ‘open’ from the side turned to the broad 3700–3000 cm^−1^ absorption band. This may indicate some information about some additional hydrogen bonds formation possibilities either the development of hydrophobic interactions which are crucial for understanding the nature of cellulose fibers. 

Some different models of intermolecular hydrogen bonding might be found in literature, e.g., 6OH⋯O2′, 2OH⋯O2′ [46]. However, the shape of the absorption band assigned to these additional hydrogen bonds is not similar to the one observed in analyzed FT-IR spectrum—the broadening of the 3700–3000 cm^−1^ to the lower wavenumber direction should be detected. However, the broadening might be evidenced as the merging of two broad absorption bands between 3700–3000 cm^−1^ and 3000–2750 cm^−1^.

Summarizing the information gathered in this section, some important and non-negligible changes in the 3000–2750 cm^−1^ peak are detected. Moreover, these variations are more intense in the region closer to the broad 3700–3000 cm^−1^ absorption band assigned to the intra- and intermolecular hydrogen bonds. Therefore, the possibility of assigning some deconvoluted signals from the region of 3000–2750 cm^−1^ to hydroxyl group-incorporated hydrogen bonding is indicated (e.g., peaks no. 1, 2 or 3 situated in the neighborhood of the broad 3700–3000 cm^−1^ absorption band assigned to hydrogen bonds). However, this requires further analysis and confirmation by different research studies.

### 3.2. Near Infrared Spectroscopy Analysis

#### 3.2.1. General Description of the Spectra

NIR experiment is known to be more fragile to the changes in polar molecules. Therefore, it was employed in this research in order to assess the variations in cellulose supramolecular structure observed via FT-IR [40]. Figure 9a describe the changes detected while recording the NIR spectra of cellulose samples which exhibit different moisture content. Moreover, in Table 2 some of the peak wavenumbers are assigned to the particular chemical groups present in the chemical structure of the cellulose fibers. 

The region marked with red color, between 7500–6000 cm^−1^, is assigned to the hydrogen bonding developed in the analyzed specimens [43]. Signals between 7200–6100 cm^−1^ are attributed to the first overtone of OH stretching vibrations. Furthermore, free −OH groups might be found at higher frequencies (6955–6970 cm^−1^) while lower wavenumbers are generally attributed to the OH bands involved in hydrogen bonds from crystalline and semi-crystalline cellulose [43]. The discussed absorption band was deconvoluted (Figure 9b–d) and furtherly described in Section 3.2.2. 

Moreover, another peak marked in Figure 9a with a grey color seems to be very interesting regarding the cellulose water content tracking. The intensity of the peak at approximately 5175 cm^−1^ gradually increases with the raising moisture content in cellulose sample [49]. 

#### 3.2.2. Deconvolution of the Peak between 7500–6000 cm^−1^ (Hydrogen Bonding, Crystalline Regions)

As it was mentioned before, the broad peak between 7500–600 cm^−1^ was resolved into 6 separate signals (Figure 9b–d) assigned to different types of −OH interactions: peak no. 1- hydroxyl groups, peak no. 2- hydroxyl groups in the amorphous region, peak no. 3- hydroxyl groups in the semi-crystalline region, peak no. 4–3OH⋯O5 (crystalline region), peak no. 5–2OH⋯O6 (crystalline region), peak no. 6- hydroxyl groups [43].

Further analysis of deconvolution results is presented in Figure 10. It could be observed that resolved signals are changing their area, wavenumber and absorbance in a less regular way in comparison with the signals observed in FT-IR spectra (Figure 6 and Figure 8). The analyzed values are not constantly raising either decreasing-they vary during the measurement going up and down depending on the moisture content in cellulose fibers. This may indicate that the order of macromolecules in cellulose sample is constantly changing during the moisture absorption process [40]. 

Moreover, it could be observed that the peak attributed to amorphous region (peak no. 2) is shifting to the higher wavenumbers, while peaks assigned to crystalline, high-ordered structures (peak no. 3–5) are shifting, in general, to the lower values, e.g., peak no. 3- from (6769 ± 7) cm^−1^ to (6749 ± 15) cm^−1^ and peak no. 5- from (6311 ± 4) cm^−1^ to (6300 ± 15) cm^−1^. 

According to the literature, crystalline peaks shift to the higher wavenumber direction due to the disintegration of crystalline structure and amorphous peak gradually shifts to the opposite direction, suggesting an increase in the population of the disordered (less interacted) structures-development of glassy/amorphous component [43]. 

This means that the changes observed for the analyzed cellulose specimens, indicate the raise of crystalline region amount during the moisture absorption which does not agree with the previous results from the FT-IR investigation. 

In the previous section it was evidenced that with the increasing moisture content, the crystalline regions amount decreased, but at the same time the raise in hydrogen bonding intensity might have been detected due to the increasing amount of water molecules in the system (additional hydrogen bonds). Moreover, water molecules penetrating entangled cellulose chains might loosen the biopolymer structure contributing to its swelling rather than increasing the amount of crystalline, high-ordered structures [4,6].

Therefore, similarly, the observed changes in NIR spectra might have been influenced by the water molecules and not the order of cellulose macromolecules. In this way, the fully precise interpretation of crystallinity changes is not possible. Cellulose-water hydrogen bonding, in case of infrared spectroscopy methods, seems to affect the spectra in the similar way as cellulose-cellulose interactions.

Furthermore, after performing the deconvolution of 7500–6000 cm^−1^ absorption band only intramolecular hydrogen bonds (Figure 10d,e) might be observed: 3OH⋯O5 (peak no. 4), 2OH⋯O6 (peak no. 5). Their parameters change a lot during the moisture absorption process and the 3OH⋯O5 seems to exhibit more regular variations throughout the investigation. The wavenumber of both peaks is shifting in the direction toward lower values. Moreover, the intensity/area of the peak no. 4 assigned to 3OH⋯O5 raises during the moisture absorption indicating the intensification of this type of interactions.

Taking into consideration the gathered data, it could be concluded that, according to NIR investigation, the order of macromolecules in cellulose sample is constantly changing during the moisture absorption process. Moreover, NIR spectra seems to be more fragile to the water presence and therefore, the precise and reliable assessment of cellulose crystallinity degree is more troublesome comparing with the FT-IR spectra analysis.

### 3.3. Comparison of Cellulose Properties before and after Moisture Absorption/Desorption

In order to summarize the observed cellulose structure changes during the moisture absorption experiment, the following comparison has been prepared. According to the fact that NIR spectra seems to be highly affected by the moisture content and it is troublesome to interpret the visible changes (whether the signals are changing only due to the cellulose-cellulose or cellulose-water hydrogen bonding), only the results from FT-IR investigation were taken into consideration.

The comparison between different states of analyzed cellulose fibers is presented: 1- cellulose after conditioning in desiccator with P_4_O_10_; 2- cellulose after 24 h of water absorption in desiccator filled with saturated solution of KNO_3_; 3- cellulose after 8 h of drying in the laboratory oven at 100 °C.

Considering Figure 11, it could be noticed that even after a one cycle of cellulose wetting-drying (regarding only moisture absorption and not full cellulose wetting) some slight changes in the biopolymer supramolecular structure might be detected. Cellulose does not necessarily come back to the native state and the primary structure is never fully regained, e.g., slight decrease in TCI value from 1.39 ± 0.3 to 1.34 ± 0.3 (Figure 11a), differences in intra- and intermolecular hydrogen bond energies and length (Figure 11d–f). 

On the other hand, regarding original and dried state, one wetting-drying cycle seems not to have a significant influence on the LOI and HBI values (Figure 11b,c). Nevertheless, it was proved that HBI is highly affected by the moisture content and water molecules presence may determine the parameter value [24]. In turn, LOI is correlated to the overall degree of order in cellulose, while TCI is referred to be proportional to the crystallinity degree of cellulose [31].

Another interesting fact is the difference between the dry and wet state, e.g., LOI slightly raises from 0.56 ± 0.02 to 0.57 ± 0.01, TCI drops from 1.39 ± 0.03 to 1.31 ± 0.04 and HBI increases from 1.31 ± 0.05 to 1.7 ± 0.1. Nonetheless, the most interesting differences are visible regarding the energy and the distance of the hydrogen bonds in cellulose fibers (Figure 11d–f).

Regarding the wet state, the 3OH⋯O5 intramolecular hydrogen bond becomes shorter from (27,786 ± 2) 10^−5^ nm to (27,770 ± 5) 10^−5^ (as the energy of the bond raises with the increasing moisture content), while the distances of 2OH⋯O6 intramolecular and 6OH⋯O3′ intermolecular hydrogen bonds decrease. This causes a specific shrinkage in the cellulose macromolecule. The model of possible changes in the bond lengths is revealed in Figure 12.

The idea of moisture effect on the 3OH⋯O5 has been proposed before in Section 3.1.2 and this phenomenon has been also indicated in another research study [35]. It was explained by the authors that the water probably affects the vibration of the mentioned hydrogen bond and, therefore, the change in the bond length could be detected. 

What should be underlined, this slight variation in intra- and intermolecular hydrogen bond distances, in turn, may affect the cellulose stress transfer and contribute to the changes in the fiber molecular packing and greatly influence natural fiber behavior which has been described in literature [1]. 

## 4. Conclusions

The presented research shows that Fourier-transform infrared and near infrared spectroscopy techniques might be successfully employed in analysis of cellulose supramolecular structure. Nevertheless, such analysis is based only on data generation and fittings. The assumption made for the study and deconvolution of data should be validated with other control characterization techniques, e.g., X-ray diffraction (XRD), solid-state nuclear magnetic resonance (NMR). 

IR methods are incredibly useful in introductory analysis of cellulose because they are quick, common and the results might be easily interpreted. Nevertheless, they are only indicative and the spectra may be randomly affected by many factors, e.g., NIR spectra can exhibit baseline shifts depending on sample preparation or porosity. Therefore, further analysis with different methods mentioned above is highly advised.

Considering the results obtained in this research, the cellulose spectra analysis may indicate various valuable information about the possible biopolymer features regarding:inter- and intramolecular hydrogen interactions development,changes in the crystallinity and amorphous regions,moisture content in the biopolymer structure with its possible influence on the energy and length of hydrogen bonds, cellulose crystallinity degree and molecular packing.

Therefore, IR methods seem to be useful while considering the quick and introductory study of cellulose supramolecular structure providing a wide range of data. Moreover, presented analysis approach may be successfully employed in assessing the biopolymer properties regarding its, e.g., chemical/physical modification and origin.

## Figures and Tables

**Figure 1 materials-13-04573-f001:**
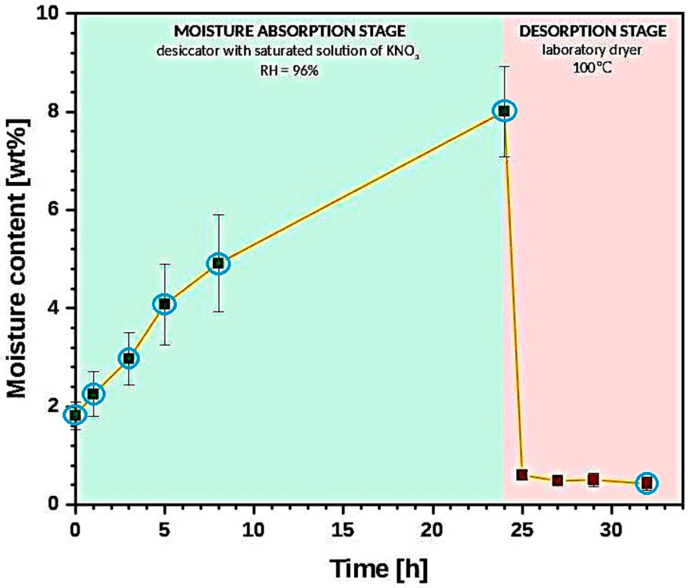
Graph illustrating the changes of moisture content throughout the carried out experiments. Points taken into consideration during the analysis of results are marked with the blue circles.

**Figure 2 materials-13-04573-f002:**
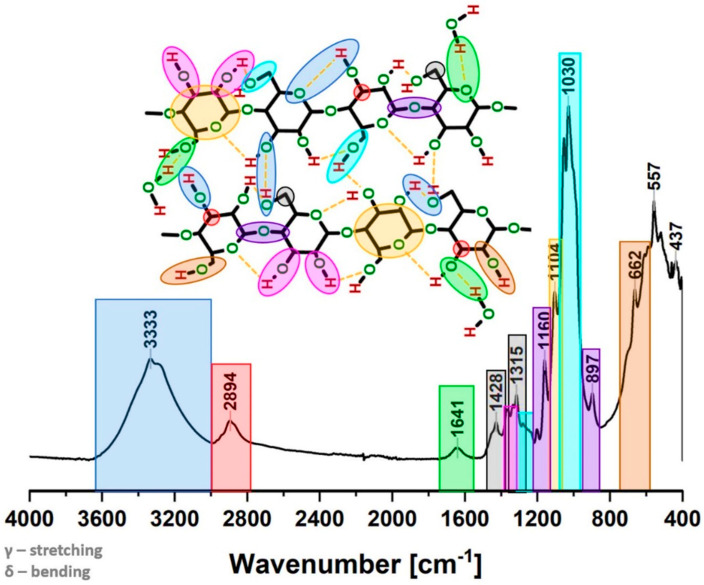
Exemplary Fourier-transform infrared spectra for a sample tested after 24 h of moisture absorption in the desiccator filled with saturated solution of KNO_3_ (RH = 96%). Chemical moieties assigned to the particular frequencies.

**Figure 3 materials-13-04573-f003:**
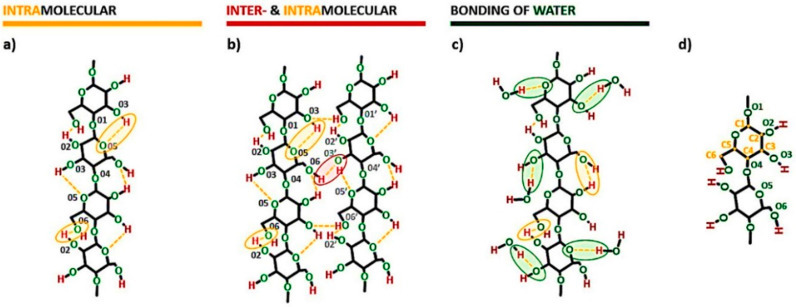
Exemplary models of hydrogen bonds in cellulose fibers: (**a**) intramolecular hydrogen bonding, (**b**) intra- and intermolecular hydrogen bonding, (**c**) cellulose-water hydrogen bonding possibilities with the description of oxygen and carbon atoms numbering (**d**). Intramolecular hydrogen bonds: 3OH⋯O5 and 2OH⋯O6. Intermolecular hydrogen bonds: 6OH⋯O3′.

**Figure 4 materials-13-04573-f004:**
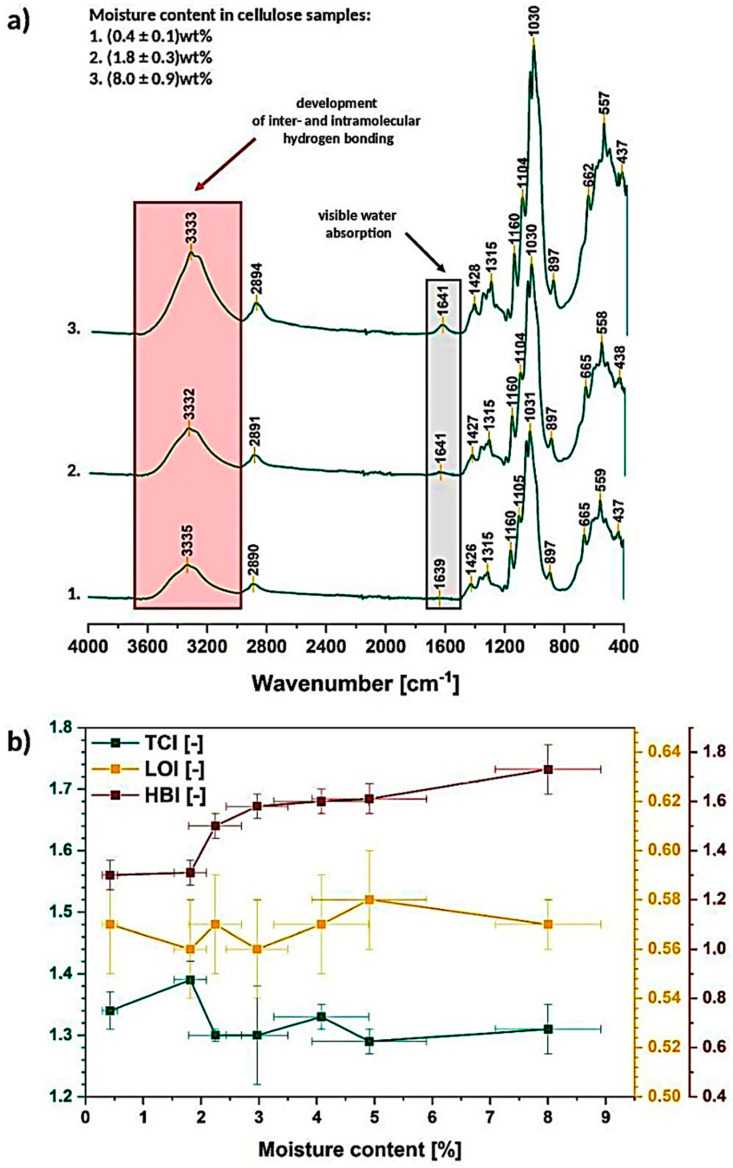
Exemplary Fourier-transformation infrared spectra of cellulose of different moisture content (**a**) and graph illustrating changes in TCI, LOI, HBI as a function of moisture content (**b**).

**Figure 5 materials-13-04573-f005:**
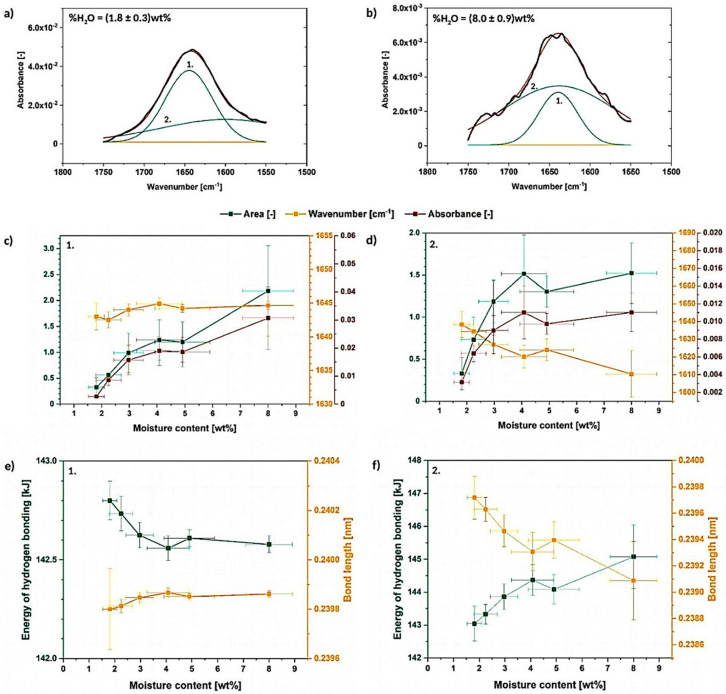
Deconvoluted absorption band between 1750–1550 cm^−1^ for the cellulose moisture content of (1.8 ± 0.3) wt% (**a**) and (8.0 ± 0.9) wt% (**b**) with graphs illustrating the changes in the deconvoluted peak area, wavenumber and absorbance (**c**,**d**), as well as energy and length of hydrogen bonds assigned to the resolved signals (**e**,**f**).

**Figure 6 materials-13-04573-f006:**
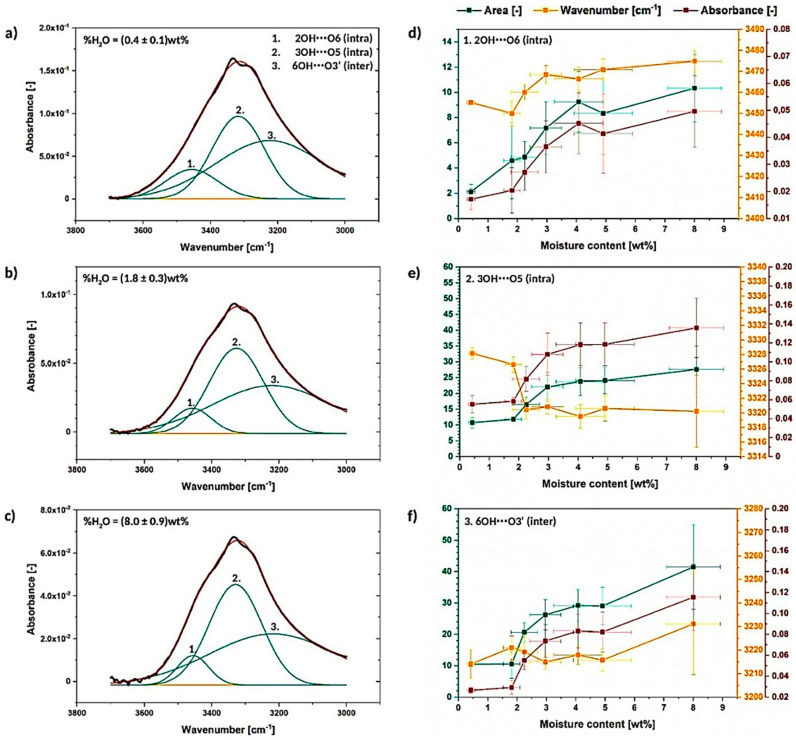
Deconvoluted absorption band between 3700–3000 cm^−1^ for the cellulose moisture content of (0.4 ± 0.1) wt% (**a**), (1.8 ± 0.3) wt% (**b**) and (8.0 ± 0.9) wt% (**c**) with graphs illustrating the changes in the deconvoluted peak area, wavenumber and absorbance (**d**–**f**).

**Figure 7 materials-13-04573-f007:**
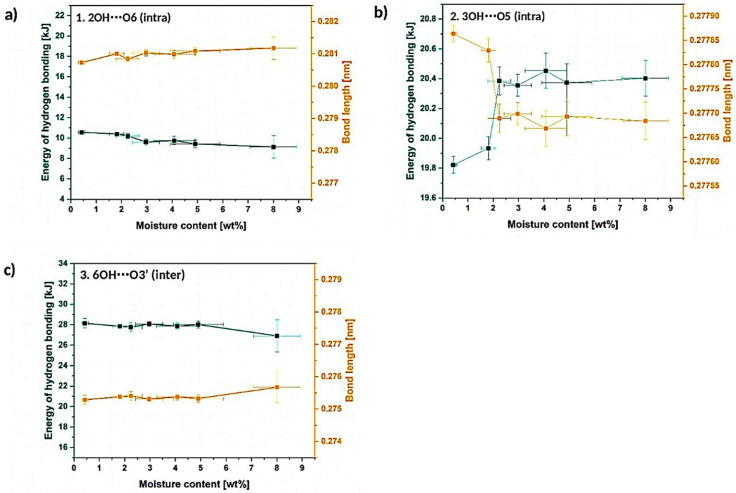
Graphs illustrating energy and length of hydrogen bonds assigned to the resolved signals: (**a**) 2OH⋯O6, (**b**) 3OH⋯O5, (**c**) 6OH⋯O3′.

**Figure 8 materials-13-04573-f008:**
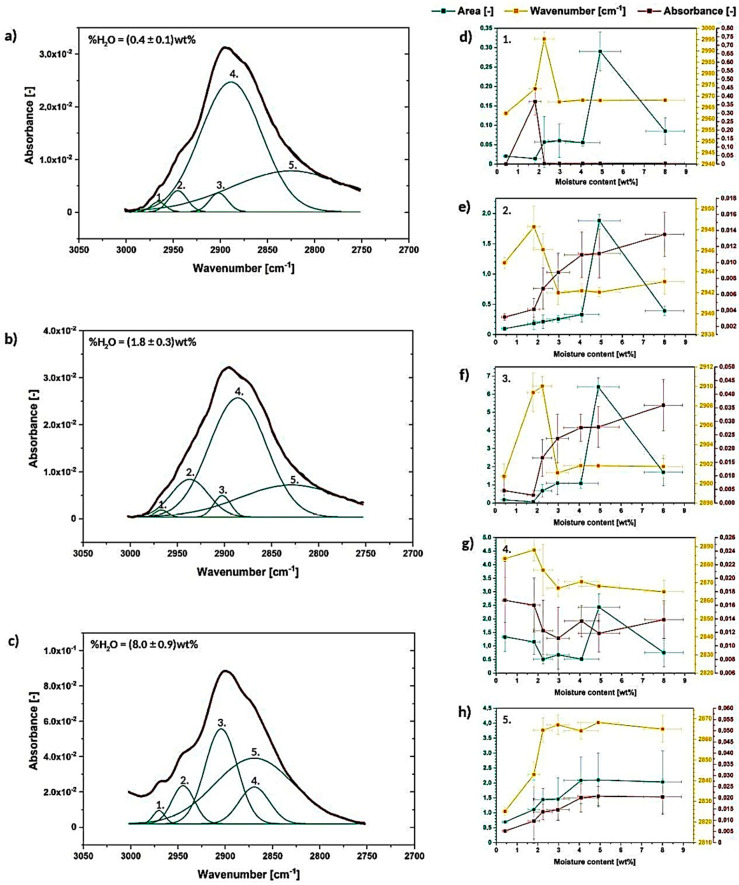
Deconvoluted absorption band between 3000–2750 cm^−1^ for the cellulose moisture content of (0.4 ± 0.1) wt% (**a**), (1.8 ± 0.3) wt% (**b**) and (8.0 ± 0.9) wt% (**c**) with graphs illustrating the changes in the deconvoluted peaks area, wavenumber and absorbance (**d**–**h**).

**Figure 9 materials-13-04573-f009:**
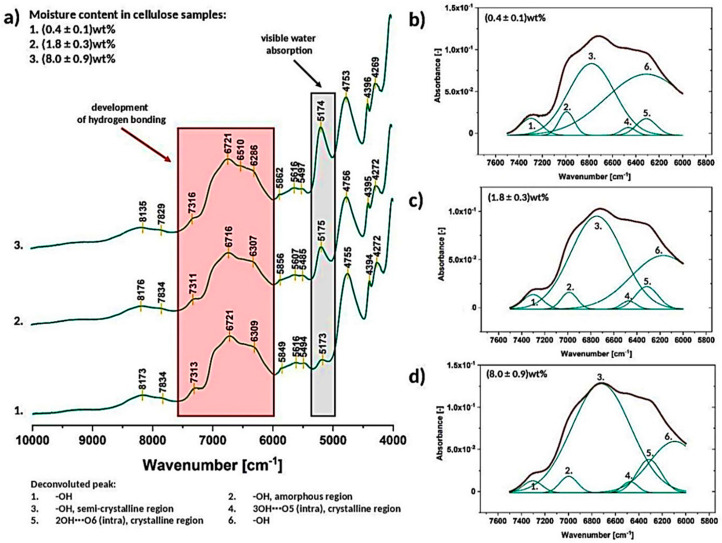
Exemplary near infrared spectra of cellulose of different moisture content (**a**) with deconvoluted absorption band between 7500–6000 cm^−1^ for the cellulose moisture content of (0.4 ± 0.1) wt% (**b**), (1.8 ± 0.3) wt% (**c**) and (8.0 ± 0.9) wt% (**d**).

**Figure 10 materials-13-04573-f010:**
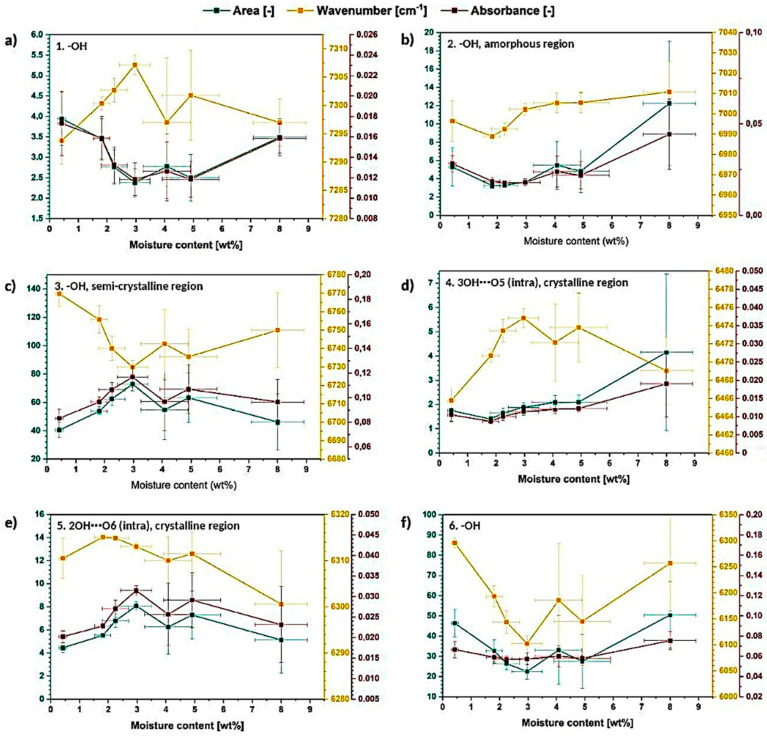
Graphs illustrating the changes in the deconvoluted peaks area, wavenumber and absorbance for the resolved signals: (**a**) −OH, (**b**) −OH and amorphous region, (**c**) −OH and semi-crystalline region, (**d**) 3OH⋯O5 and crystalline region, (**e**) 2OH⋯O6 and crystalline region, (**f**) −OH.

**Figure 11 materials-13-04573-f011:**
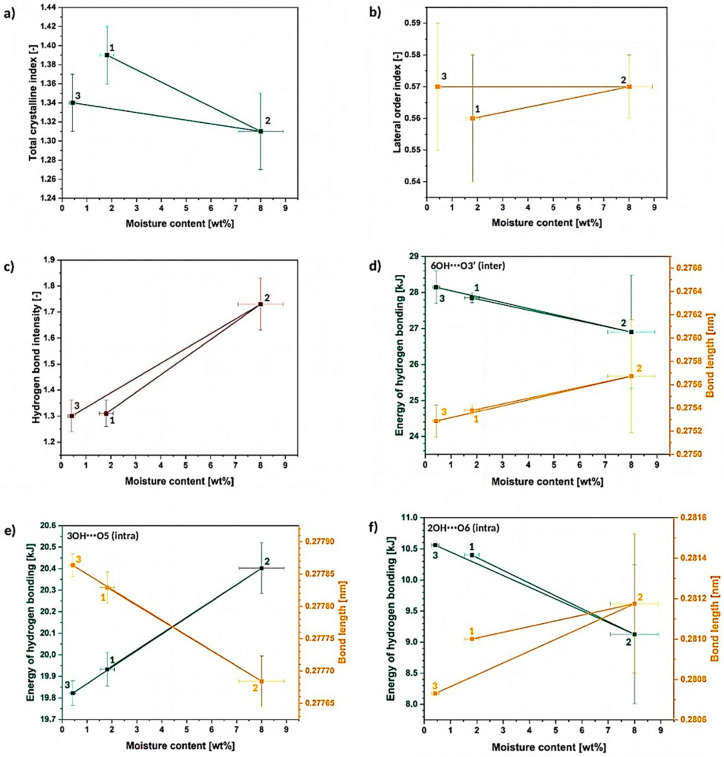
Differences between the different cellulose states: 1. before the experiment, 2. after 24 h of moisture absorption and 3. dried cellulose, regarding the changes in: (**a**) TCI, (**b**) LOI, (**c**) HBI, as well as variations in the energy and length of hydrogen bonds: (**d**) 6OH⋯O3′, (**e**) 3OH⋯O5, (**f**) 2OH⋯O6. hh.

**Figure 12 materials-13-04573-f012:**
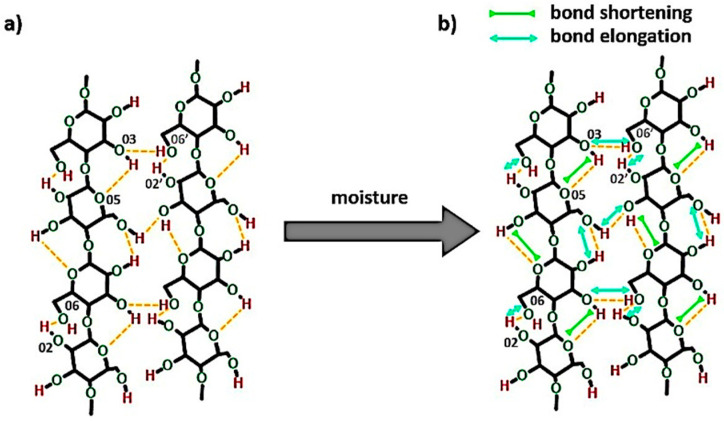
The scheme of the possible changes in the bond lengths while cellulose is subjected to the humid atmosphere (RH = 96%): (**a**) system of hydrogen bonds in the dry cellulose fibers, (**b**) system of hydrogen bonds in the cellulose of (8.0 ± 0.9) wt% moisture content.

**Table 1 materials-13-04573-t001:** Tabularized values of wavenumber attributed to the chemical moieties.

Wavenumber [cm^−1^]	Chemical Group	Ref.
3333	γOH covalent bond, hydrogen bonding	[24]
2894	γCH	[33]
1641	absorbed water (hydrogen-bonded)	[32]
1428	δCH_2_ (symmetric) at C-6; crystalline region	[32]
1372	δCH	[34]
1340	δCOH in plane at C-2 and C-3	[35]
1315	δCH_2_ (wagging) at C-6	[31]
1236	δCOH in plane at C-6	[32]
1160	γCOC at β-glycosidic linkage	[32]
1104	γ ring in plane	[36]
1030	γCO at C-6	[37]
897	γCOC at β-glycosidic linkage; amorphous region	[32]
662	δCOH out of plane	[37]

**Table 2 materials-13-04573-t002:** Tabularized values of wavenumber attributed to the chemical groups.

Wavenumber [cm^−1^]	Chemical Group	Ref.
4753	−OH, C=O	[47]
5180–5150	−OH, water	[48]
7500–6000	−OH, water, hydrogen bonds	[49]
8200–8100	C-H	[48]

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
