# Peer review of "IR Study on Cellulose with the Varied Moisture Contents: Insight into the Supramolecular Structure"

_materials, 2020, doi:10.3390/ma13204573_

Round 1

Reviewer 1 Report

The article aims at evaluating the supramolecular structure of cellulose by concentrating on the hydrogen bonds formation as a function of the moisture content. Based on the NIR results the total crystalline index, lateral order index and hydrogen bonding intensity were calculated. While the study in itself is rigorous, the cross-validation of the results with other analytical techniques would be more valuable to have conclusive statements. For instance crystalline index can be validated with the X-ray diffraction (XRD) patterns and solid-state 13C nuclear magnetic resonance (NMR). Also the discussions should be made in a way that the information gathered with NIR results are indicative as the accuracy and precision of the measurements and subsequent analysis can vary. 

More detailed comments:

It is nice that authors have developed NIR as a tool to analyze physical properties of cellulose, but the analysis are based on data generation and deconvolution/fittings. The assumptions made for the study and deconvolution/fitting of data should be validated with other control characterization techniques such as Xray-Diffraction or Solid-State NMR. When there would be a clear agreement between the properties obtained with control characterization methods and data generated with NIR and subsequent deconvolution/fitting techniques, then only such method can be validated to be conclusive. I think it is important because one of the features of NIR spectra is that they often exhibit baseline shifts, i.e. spectra shown an additive effect along the absorption axis and it varies due to several factors during sample preparation and analysis. For instance, the porosity of cellulose fibers could alter the results. Already, as shown in Figure 1, the moisture content and standard deviation of samples are overlapping, which could generate varying results. Hence, I think the analysis in this manuscript should either be presented as guideline, but not conclusive statements, or should be supported by complementary characterization methods. These changes require major revision of the manuscript.

In its current form, I do not see authors fulfilling the aim of the precise and detailed investigation of the supramolecular structure of cellulose, as stated.

Author Response

Institute of Polymer and Dye Technology

Technical University of Lodz

90-924 Lodz, ul Stefanowskiego 12/16, Poland

Tel.: +48 42 631 32 23, Fax: +48 42 636 25 43

October 5, 2020

Materials

Dear Professor,

We are resubmitting our revised paper entitled IR Study on Cellulose with the Varied Moisture Contents: Insight into the Supramolecular Structure by, Stefan Cichosz, Anna Masek with a request to reconsider it for publication in Materials.

We have carefully considered the Editor and Reviewers' comments. The manuscript was revised exactly according to these comments. The list of responses to the reviewer’s comments and corrections made in the manuscript is attached.

The manuscript has not been previously published, is not currently submitted for review to any other journal, and will not be submitted elsewhere before a decision is made by this journal.

For correspondence please use the following information:

corresponding author: Anna Masek

Institute of Polymer and Dye Technology

Technical University of Lodz

90-924 Lodz, ul Stefanowskiego 12/16, Poland

Tel.: +48 42 631 32 93

Fax: +48 42 636 25 43

e-mail: anna.masek@p.lodz.pl

Yours sincerely,

Ph. D., D.Sc. Anna Masek

All changes are marked with a green colour through whole manuscript.

Reviewer #1

The article aims at evaluating the supramolecular structure of cellulose by concentrating on the hydrogen bonds formation as a function of the moisture content. Based on the NIR results the total crystalline index, lateral order index and hydrogen bonding intensity were calculated. While the study in itself is rigorous, the cross-validation of the results with other analytical techniques would be more valuable to have conclusive statements. For instance crystalline index can be validated with the X-ray diffraction (XRD) patterns and solid-state 13C nuclear magnetic resonance (NMR). Also the discussions should be made in a way that the information gathered with NIR results are indicative as the accuracy and precision of the measurements and subsequent analysis can vary.  

More detailed comments: 

It is nice that authors have developed NIR as a tool to analyze physical properties of cellulose, but the analysis are based on data generation and deconvolution/fittings. The assumptions made for the study and deconvolution/fitting of data should be validated with other control characterization techniques such as Xray-Diffraction or Solid-State NMR. When there would be a clear agreement between the properties obtained with control characterization methods and data generated with NIR and subsequent deconvolution/fitting techniques, then only such method can be validated to be conclusive. I think it is important because one of the features of NIR spectra is that they often exhibit baseline shifts, i.e. spectra shown an additive effect along the absorption axis and it varies due to several factors during sample preparation and analysis. For instance, the porosity of cellulose fibers could alter the results. Already, as shown in Figure 1, the moisture content and standard deviation of samples are overlapping, which could generate varying results. Hence, I think the analysis in this manuscript should either be presented as guideline, but not conclusive statements, or should be supported by complementary characterization methods. These changes require major revision of the manuscript. 

In its current form, I do not see authors fulfilling the aim of the precise and detailed investigation of the supramolecular structure of cellulose, as stated. 

Answer: We are incredibly grateful for this comment as it draws our attention to a few problems concerning the carried out research and the submitted manuscript. We agree that the article in the previous form might have been misleading in its narration. Nonetheless, we would like to underline, it was never our aim to claim that the IR study is an absolute method for cellulose supramolecular structure analysis. We also agree that the submitted manuscript is some kind of the guideline of how to investigate the FT-IR/NIR spectra regarding the changes in cellulosic materials at the supramolecular level. 

Therefore, the abstract, conclusions and the narration of whole article has been altered in order to fulfill the Revier’s requirements and stay true to the reader. Additionally, the good correspondence of the FT-IR results to different complementary methods has been presented before in literature which has been mentioned in the manuscript in a few places, e.g.: Moreover, it is highly advised to be familiar with the crystalline and amorphous regions spectral displacement as the ratio between the peaks assigned to these two cellulose states, namely 1430-1420/897 cm-1, may indicate some information about the crystallinity degree of cellulose and is referred as lateral order index (LOI) [31]. Additionally, the ratio between the peaks at ~1370 cm-1 and 2900-2890 cm-1 is regarded as total crystalline index (TCI) [31]. Both of these parameters are very useful in cellulose properties description and reliably correspond with the crystalline index calculated according to the X-ray diffraction (XRD) experiment results [32].  

We are thankful for this inspiring and precise comment. 

Reviewer 2 Report

The manuscript materials-953332 demonstrates the development of the method for studying intermolecular and intramolecular hydrogen bonds in cellulose materials and in my opinion can be published in Materials. However, I think that the manuscript can be improved for reading. This will increase interest for a wider range of readers. Here are some suggestions:

  1. Abstract should show the essence of the article, in my opinion, and not be only a summary of what the authors did. It is not clear what the achievements of the authors are compared with the known data. What is fundamentally decided, what effect is achieved. Abstract must be corrected.
  2. Figure 1: The need to show the same graph three times is not clear. In its current form, Figure 1 is not clear and its informative value for the reader is not obvious.
  3. Figure 2: Some shades of color are poorly selected. For some bands it is not clear to which bonds they correspond. And where is the band absorption of the covalent H – O bond?
  4. Table 1: Where is the band absorption of the covalent H – O bond?
  5. Figure 8, lines 434-439: Why are 5 absorption maxima chosen? What does each maximum refer to?
  6. Conclusion needs to be improved. Conclusion should show the essence of the article, in my opinion, and not be only a summary of what the authors did. It is not clear what the achievements of the authors are compared with the known data. What is fundamentally decided, what effect is achieved.

Author Response

Institute of Polymer and Dye Technology

Technical University of Lodz

90-924 Lodz, ul Stefanowskiego 12/16, Poland

Tel.: +48 42 631 32 23, Fax: +48 42 636 25 43

October 5, 2020

Materials

Dear Professor,

We are resubmitting our revised paper entitled IR Study on Cellulose with the Varied Moisture Contents: Insight into the Supramolecular Structure by, Stefan Cichosz, Anna Masek with a request to reconsider it for publication in Materials.

We have carefully considered the Editor and Reviewers' comments. The manuscript was revised exactly according to these comments. The list of responses to the reviewer’s comments and corrections made in the manuscript is attached.

The manuscript has not been previously published, is not currently submitted for review to any other journal, and will not be submitted elsewhere before a decision is made by this journal.

For correspondence please use the following information:

corresponding author: Anna Masek

Institute of Polymer and Dye Technology

Technical University of Lodz

90-924 Lodz, ul Stefanowskiego 12/16, Poland

Tel.: +48 42 631 32 93

Fax: +48 42 636 25 43

e-mail: anna.masek@p.lodz.pl

Yours sincerely,

Ph. D., D.Sc. Anna Masek

All changes are marked with a green colour through whole manuscript.

Reviewer #1

Reviewer #2

The manuscript materials-953332 demonstrates the development of the method for studying intermolecular and intramolecular hydrogen bonds in cellulose materials and in my opinion can be published in Materials. However, I think that the manuscript can be improved for reading. This will increase interest for a wider range of readers.

The comments are listed below:

  1. Abstract should show the essence of the article, in my opinion, and not be only a summary of what the authors did. It is not clear what the achievements of the authors are compared with the known data. What is fundamentally decided, what effect is achieved. Abstract must be corrected.

Answer: We are thankful for this comment and, therefore, the abstract has been corrected and improved in order to make it clear and interesting: The following article is the first attempt to investigate the supramolecular structure of cellulose with the varied moisture content by the means of Fourier-transform and near infrared spectroscopy techniques. Moreover, authors aimed at the detailed and precise presentation of IR spectra interpretation approach in order to create a reliable guideline for other researchers. On the basis of obtained data, factors indicating biopolymer crystallinity and development of hydrogen interactions were calculated and the peaks representing hydrogen bonding (7500-6000 cm-1, 3700-3000 cm-1, and 1750-1550 cm-1) were resolved using the Gaussian distribution function. Then, the deconvoluted signals have been assigned to the specific interactions occurring at the supramolecular level and the hydrogen bond length, as well bonding-energy were established. Furthermore, not only was the water molecules adsorption observed, but also the possibility of the 3OHO5 intramolecular hydrogen bond shortening in the wet state was found – from (27786 ± 2) ∙ 10-5 nm to (27770 ± 5) ∙ 10-5 nm. Additionally, it was proposed that some deconvoluted signals from the region of 3000-2750 cm-1 might be assigned to the hydroxyl group-incorporated hydrogen bonding, which is, undoubtedly, a scientific novelty as the peak was not resolved before.

  1. Figure 1: The need to show the same graph three times is not clear. In its current form, Figure 1 is not clear and its informative value for the reader is not obvious.

Answer: We agree with this comment and Figure 1 was altered. Only one graph with marked points is presented.

  1. Figure 2: Some shades of color are poorly selected. For some bands it is not clear to which bonds they correspond. And where is the band absorption of the covalent H – O bond?

Answer: We are thankful for drawing attention to this problem. The colours has been changed and bonds reassigned to the appropriate chemical moieties.

  1. Table 1: Where is the band absorption of the covalent H – O bond?

Answer: We are sorry for this mistake. Table 1 was improved according to the comment.

  1. Figure 8, lines 434-439: Why are 5 absorption maxima chosen? What does each maximum refer to?

Answer: The peak between 3000-2750 cm-1 was never before deconvoluted. It has been divided into 5 different signals as in the wet state (Fig. 8c) 4 or 5 peaks could be visible. Nevertheless, the Gaussian distribution fitting could have not been performed in case of deconvolution into 4 peaks. 5 signals is the lowest number of peaks into which the wide absorption band at 3000-2750 cm-1 may be divided. This information has been incorporated into the manuscript in section 3.1.3.: Furthermore, an interesting case is revealed in Fig. 8. In literature it is often underlined that the peak between 3000-2750 cm-1 might be highly affected by the water content in cellulose sample. Nevertheless, it was never deconvoluted into separate signals.

Some studies assign this absorption band only to C-H bonds typical to alkyl chains [42,44] and only in the researches which debates about the moisture content in natural fibers, it is partially referred to be influenced by water content [24,32,34]. Moreover, an interesting paper from 1998 indicates the possibility of assigning one of the deconvoluted signals from this broad absorption band to -OH moieties [45].

Regarding the performed calculations, the absorption band between 3000-2750 cm-1 could be resolved into 5 separate signals (Fig. 8a-c). It is the lowest number of peaks into which the broad absorption band might be successfully divided. Nevertheless, according to the current state of knowledge, it is troublesome to assign these deconvoluted signals to some particular chemical bonds present in the structure of cellulose. However, possibly, most of them could be attributed to different C-H bonds which are present in the structure of natural fiber [42,44].

  1. Conclusion needs to be improved. Conclusion should show the essence of the article, in my opinion, and not be only a summary of what the authors did. It is not clear what the achievements of the authors are compared with the known data. What is fundamentally decided, what effect is achieved.

Answer: We think this is a very valuable comment. Conclusion has been corrected:
The presented research shows that Fourier-transform infrared and near infrared spectroscopy techniques might be successfully employed in analysis of cellulose supramolecular structure. Nevertheless, such analysis is based only on data generation and fittings. The assumption made for the study and deconvolution of data should be validated with other control characterization techniques, e.g., X-ray diffraction (XRD), solid-state nuclear magnetic resonance (NMR).

IR methods are incredibly useful in introductory analysis of cellulose because they are quick, common and the results might be easily interpreted. Nevertheless, they are only indicative and the spectra may be randomly affected by many factors, e.g., NIR spectra can exhibit baseline shifts depending on sample preparation or porosity. Therefore, further analysis with different methods mentioned above is highly advised.

Considering the results obtained in this research, the cellulose spectra analysis may indicate various valuable information about the possible biopolymer features regarding:

  • inter- and intramolecular hydrogen interactions development,
  • changes in the crystallinity and amorphous regions,
  • moisture content in the biopolymer structure with its possible influence on the energy and length of hydrogen bonds, cellulose crystallinity degree and molecular packing.

Therefore, IR methods seem to be useful while considering the quick and introductory study of cellulose supramolecular structure providing a wide range of data. Moreover, presented analysis approach may be successfully employed in assessing the biopolymer properties regarding its, e.g., chemical/physical modification and origin.

Round 2

Reviewer 1 Report

Authors have significantly modified the manuscript, and with revision the abstract and conclusions are now true to the readers. Clearly, the manuscript now discusses the significance of NIR methods as an indicative tools for cellulose characterization and highlights its limitation. Hence based on the changes, I can recommend the revised manuscript for publication in Materials.

Reviewer 2 Report

The second version of the manuscript materials-953332  can be published in present form.